# Spatial Uncertainty in Modeling Inhalation Exposure to Volatile Organic Compounds in Response to the Application of Consumer Spray Products

**DOI:** 10.3390/ijerph18105334

**Published:** 2021-05-17

**Authors:** Yerin Jung, Yoonsub Kim, Hwi-Soo Seol, Jong-Hyeon Lee, Jung-Hwan Kwon

**Affiliations:** 1Division of Environmental Science and Ecological Engineering, Korea University, 145 Anam-ro, Seongbuk-gu, Seoul 02841, Korea; yerinj3@uci.edu (Y.J.); kys0437@korea.ac.kr (Y.K.); 2Program in Public Health, University of California, Irvine, CA 92697, USA; 3Environment & Safety Research Center, Samsung Electronics Co. Ltd., Samsungjeonja-ro 1, Gyeonggi-do, Hwaseong-si 18448, Korea; 4EH R&C, Environmental Research Center, 410 Jeongseojin-ro, Seo-gu, Incheon 22689, Korea; hs.seol@ehrnc.com (H.-S.S.); jhleecheju@gmail.com (J.-H.L.)

**Keywords:** micro environmental modeling, volatile organic compounds (VOCs), inhalation exposure, spray product, proton transfer reaction mass spectrometry (PTR-MS)

## Abstract

(1) Background: Mathematical exposure modeling of volatile organic compounds (VOCs) in consumer spray products mostly assumes instantaneous mixing in a room. This well-mixed assumption may result in the uncertainty of exposure estimation in terms of spatial resolution. As the inhalation exposure to chemicals from consumer spray products may depend on the spatial heterogeneity, the degree of uncertainty of a well-mixed assumption should be evaluated under specific exposure scenarios. (2) Methods: A room for simulation was divided into eight compartments to simulate inhalation exposure to an ethanol trigger and a propellant product. Real-time measurements of the atmospheric concentration in a room-sized chamber by proton transfer reaction mass spectrometry were compared with mathematical modeling to evaluate the non-homogeneous distribution of chemicals after their application. (3) Results: The well-mixed model overestimated short-term exposure, particularly under the trigger spray scenario. The uncertainty regarding the different chemical proportions in the trigger did not significantly vary in this study. (4) Conclusions: Inhalation exposure to aerosol generating sprays should consider the spatial uncertainty in terms of the estimation of short-term exposure.

## 1. Introduction

Volatile organic compounds (VOCs) are commonly used in consumer spray products for various purposes, such as personal care and biocides [1,2,3]. Estimating the exposure to VOCs is important for their safe use, and some VOCs in products are known to be harmful when inhaled. Indoor VOCs, including formaldehyde, aromatics, and aliphatics, are known to exhibit positive correlations with respiratory or immune diseases in children [4]. Increase of oxidative stress markers in murine brain and damage of learning and memory functions of mice by exposure to VOCs was observed and implies potential neurobehavioral disturbance by VOCs [5]. Additionally, the frequent use of spray products containing volatile disinfectants could potentially result in the development of health-related effects, such as adult asthma [2,6]. Children living nearby petrochemical plant emitting VOCs in Argentina were observed to have more asthma and respiratory symptoms compared to the control [7].

Exposure assessment of these products has been requested by several regulatory authorities for the product’s evaluation and authorization [8]. As the chemical analysis of active ingredients for the exposure assessment can only be conducted for certain cases due to time and cost, mathematical modeling is commonly employed for VOCs in such products [3,9]. Models for the exposure assessment are divided into two parts: physical models for estimating the chemical concentration and exposure models for simulating the activity of a person [10,11]. Most physical models for assessing the personal exposure to consumer products (i.e., ConsExpo [9]) are based on mass-balance equations in a microenvironment where instantaneous mixing is assumed [12]. Depending on the conditions at which the consumer spray products are used, this simplifying assumption often fails to predict the exposure concentration [13]. For example, a study on hotel housekeepers attached with personal samplers reported nearly double the amount of VOC exposure compared to that calculated based on the concentration of the room assuming complex mixing [14]. This deviation could be explained by the heterogeneity of the ambient concentration during the use of consumer products. Computational fluid dynamics (CFD) techniques, which divide a space into extremely fine grid cells, can be used to predict the precise spatial distribution of target chemicals from spray products [15]. Despite providing detailed information on fine spatial scales, CFD-applied models are not effective for generic population exposure modeling owing to their high cost and demanding inputs such as the heat transfer coefficient, room scale, supplied air velocity, and occupant behavior [12,15].

Microenvironmental models were thus suggested to supplement the mass-balance models based on the well-mixed assumption but were adjusted to reduce the number of microenvironments to avoid complexity, such as in CFD techniques. The near field-far field (NF-FF) model divides a room into two well-mixed zones: a near-field where the emission source is placed and a far-field where the air exchange occurs with the near-field zone and the outdoor air [16]. Exposure estimates based on the NF-FF model agreed well with the measurement data; however, greater uncertainty between the actual measurement and modeling data is inevitable unless the input parameters are specified in experiments [17].

Thus, the uncertainty of inhalation exposure should be considered when investigating the spatial resolution for the better application of models based on well-mixed assumptions. In this study, a model dividing an entire room into eight microenvironments (henceforth referred to as compartments) was used to estimate the uncertainty of prevalent models assuming well-mixed conditions. Exposure scenarios using a VOC-containing spray product were simulated in a room-sized chamber, considering the factors contributing to the spatial uncertainty, i.e., the type of spray and volume fraction of VOCs in the solution. Spatial uncertainty was monitored real-time in each compartment using a proton-transfer-reaction quadrupole mass spectrometer (PTR-qMS). The concentration deviation in each compartment was compared to evaluate the spatial uncertainty of the measured and estimated exposure concentrations, and factors contributing to uncertainty in the scenarios were evaluated for future use.

## 2. Materials and Methods

### 2.1. Materials

Ethanol (HPLC solvent grade) was purchased from Sigma-Aldrich (St. Louis, MO, USA). An empty trigger-type spray was used for experiments with a known ethanol emission amount. A dust cleaner containing *n*-butane was purchased from a local market in Seoul, Korea.

### 2.2. Test Chambers

Changes in the ambient air concentration of test VOCs were measured in real-time using a custom-made chamber [18] (Appendix A). A 30 m^3^ stainless room-sized chamber was manufactured to meet the criteria set by the American Society for Testing and Materials (ASTM), and United States Environmental Protection Agency (U.S. EPA) [19,20]. Relative humidity (range: 38–41%) and temperature (range: 21.5–23.6 °C) were adjusted using a carbon and high efficiency particulate air (HEPA) filter, and the air change rate (ACR; λ, h^−1^) of the chamber was adjusted to 1.0 h^−1^ using a heating, ventilation, and air conditioning (HVAC) system.

A small custom-made acrylic chamber was used to calibrate the PTR-qMS using a standard chemical (Appendix A). The detailed description of the PTR-qMS is found in Section 2.5. The volume of the acrylic chamber was 0.29 m^3^, and a fan was used to homogeneously mix the evaporated chemical.

### 2.3. Exposure Scenarios

One of the default exposure scenarios offered by ConsExpo Web (version 1.0.5) was chosen as a representative commercially available VOC-containing spray product in Seoul, Korea. The surrogate VOCs for evaluating the uncertainty were selected for the experiment considering the ease of real-time monitoring using a PTR-qMS. In this study, the scenario using an all-purpose cleaning spray was selected and divided into two, a trigger-type spray emitting mostly liquid aerosols (exposure scenario 1) and a propellant-type spray emitting gas (exposure scenario 2). In exposure scenario 1, a trigger-type spray containing an ethanol/water mixture at 20, 50, and 80% (*v*/*v*) was used to evaluate the effects of solution composition on the spatial uncertainty. For a propellant-type spray in exposure scenario 2, a commercial dust cleaner containing *n*-butane was used, as almost no aerosols were detected using an optical particle sizer (OPS 3330; TSI Incorporated, Shoreview, MN, USA) in a room-sized chamber. Size distributions of aerosol from the application of the trigger-type and the propellant-type spray are depicted in Appendix A.

### 2.4. Modeling

Two models with different spatial resolutions were used to estimate the atmospheric concentration of VOCs in a test chamber. In model 1, instantaneous and complete mixing in the whole chamber after the application of the products was assumed (Figure 1a). Conversely, the test chamber was divided into eight completely mixed air compartments in model 2, as shown in Figure 1b,c.

Assuming that VOC losses are dominated by ventilation and their concentration in the infiltrating air is negligible, the mass-balance equation for model 1 is expressed as
(1)$$V\frac{dC_{air}}{dt}=-QC_{air}+R_{em}$$
where *V* is the room volume (m^3^), *C_air_* is the atmospheric concentration of the target VOC (mg m^−3^), *t* is the time (h), *Q* is the air flow rate to the chamber (m^3^ h^−1^; *Q = V·λ*), and *R_em_* is the emission rate of the VOC from the product (mg h^−1^).

In model 2, the product is applied to the source compartment, 111, while the other seven compartments contain no sources. The notation of the compartments is numbered according to the xyz-coordination (x = 1,2; y = 1,2; z = 1,2). It was assumed that the overall air flows in one direction and the concentration of compartments at y = 2 is affected by one-fourth of λ compared to that in its upstream compartment (y = 1). The air exchange rate between two adjacent compartments (*Q_ex_* [m^3^ h^−1^]; $Q_{ex}=\frac{V}{8}\cdot \lambda_{ex}$) are all equal and independent from λ. Model 2 considered the vertical air exchange between adjacent compartments as well as the horizontal air exchange. The chemical concentration in each compartment was estimated using the following equation:(2)$$\frac{V}{8}\cdot \frac{dC_{i}}{dt}=-\frac{Q}{4}C_{i}+Q_{ex}\underset{j\;=\;neighbor\;of\;i}{\sum }\left(C_{j}-C_{i}\right)+\frac{Q}{4}C_{k\;=\;upstream\;of\;i}+R_{em,i}$$
where *C_i_* is the atmospheric concentration of the target VOC in a compartment (mg m^−3^), *C_j_* and *C_k_* are three neighboring compartments and one upstream compartment, respectively, and *R_em,i_* is the emission rate in compartment *i* (mg h^−1^).

Differential equations for the compartments were simultaneously solved in R-Studio with the deSolve package, and parameter *λ_ex_* was fitted by each experimental data using the Leveberg-Marquardt nonlinear least-squares algorithm with the “minpack.lm” package [21,22].

### 2.5. Real-Time Monitoring of VOCs in the Test Chamber

Spatial variation of VOCs after the use of selected consumer products was evaluated by real-time monitoring using a PTR-qMS (Ionicon GmbH, Innsbruck, Austria). PTR-qMS was used for the detection of ethanol and PTR-qMS combined with switchable reagent ions (SRI) was used for the detection of *n*-butane, owing to the difference in sensitivity [23,24]. PTR-qMS measures real-time concentration of VOCs by monitoring ions produced from reactions of primer ions (H_3_O^+^ and O_2_^+^, in this study) with VOCs in a drift tube [25]. The temperature of the drift tube was maintained at 80 °C and its operating conditions were 2.3 mbar (p-Drift) and 600 V (U_drift_) to maintain the electric field per gas number density (E/N) at 136 Td [1 Td = 10^−17^ V cm^2^ molecule^−1^]. The current of the ion source (Ihc) was 4.0 mA with H_2_O flow at 6 sccm (standard cubic centimeter per minute). The ethanol concentration was monitored at *m*/*z* 21.00 and 47.00 for H_3_O^+^ (primary ion) and C_2_H_5_OH·H^+^, respectively. The *n*-butane concentration was monitored at *m*/*z* 34.00 and 57.00 for isotopic O_2_^+^ (primary ion) and C_4_H_9_^+^, respectively. The scan speed of each mass was 100 ms.

Each spray product was positioned at a height of 1.1 m from the floor and placed parallel to the direction of air flow along the x-axis of the chamber. The weight of the product was measured before and after application. The chamber was considered to consist of eight compartments of the equal volume. The center of each compartment was designated as a sampling point, and its distance from the nearest and farthest compartment is 1.2 m and 1.95 m, respectively. Concentrations of VOCs in eight compartments of the room chamber were monitored triplicate for the exposure scenario 1 and once for the exposure scenario 2. The concentration of ethanol or *n*-butane was measured for 20 min after the application of the product at eight sampling points, as described in Figure 1b,c. The chamber was sufficiently ventilated between tests to maintain the background concentration at least for 5 min before the application lower than 10 mg/m^3^ and 1 mg/m^3^ for exposure scenario 1 and 2, respectively. An inlet line fixed at the desired sampling point was used to send air to a PTR instrument located outside the chamber. The experiments were repeated in all eight compartments to evaluate spatial variability.

The instrument was calibrated using a standard chemical in the acryl chamber. Ethanol was injected into the chamber attached to a running fan, and the concentration was measured after no liquid was observed on the floor. The ambient concentration was calculated assuming complete evaporation and negligible losses; a good linearity was observed in the experimental range. The observed concentration by PTR-qMS was then calibrated and subtracted by each background concentration measured before the application. Next, it was smoothened with a running average of 2.5 min to decrease the instrumental noise [26]. The measured concentrations in different experiments were normalized based on the weight of the product applied to compare the independent measurements. The 0.1- and 0.3-h time-weighted averages (TWA) were calculated based on both the measured and modeled concentration data in each exposure scenario.

The coefficient of variation (CV) was used as an indicator for the spatial uncertainty of different scenarios over time. CV was calculated based on the standard deviation of the concentration of eight compartments divided by their normalized mean concentration at time *t*.

## 3. Results

Exposure scenario 1 with a trigger-type spray was simulated in a room-sized chamber, and the ethanol concentration was measured in triplicate at eight sampling points in the chamber (Figure 2). After a certain amount of time following the application (0.15 to 0.25 h), the measured values in each compartment deviated less from the well-mixed estimation (bold line). For the short-term exposure less than 0.10 h, model 1 overestimated the concentration, except in *C*_111_ and *C*_121_. As the duration of exposure was 20 min, the concentrations in all compartments converged to the predicted value within the range of 15 to 30 mg m^−3^.

Temporal variation of CV in modeling the eight compartments was estimated using different *λ_ex_* values (Figure 3). As *λ_ex_* increases (e.g., when *λ_ex_* is 10 times larger than *λ*), it is plausible to say that mixing occurs rigorously between compartments. Based on the real-time concentrations in the eight compartments shown in Figure 2, the trimmed average of the best-fitted *λ_ex_* values was 4.64 h^−1^. Although the *λ_ex_* value fits well with the modeled concentration, the spatial uncertainty of the real-time concentration is larger than that estimated using the mathematical prediction when *λ_ex_* is 4.64 h^−1^ (Figure 3).

The *n*-butane concentration was measured in the room chamber in the simulated exposure scenario 2, in which a propellant spray containing *n*-butane is employed (Figure 4a). Despite fluctuations in the concentration of the short-term exposure, the concentration in each compartment eventually reached a similar level, ranging between 0.05 and 0.15 mg m^−3^. Figure 4b shows the decreasing trends in CV values over time in two exposure scenarios. At shorter exposure times (<0.10 h), the spatial uncertainty is considerably larger in scenario 1 than that in scenario 2, but the discrepancy between the two exposure scenarios is negligible subsequently.

The composition of the VOC in the product is another factor that might influence the spatial uncertainty of the model. To test the effects of the ethanol fraction in the solution on the spatial uncertainty, three different mixing ratios of ethanol in a product were tested under exposure scenario 1 using a trigger spray. Figure 5a–c shows the real-time concentration after applying each solution. The concentration in each compartment reached a level similar to that predicted by model 1 in all three solutions, except for *C*_111_ with 80% ethanol solution spray. The decreasing trend of uncertainty upon using different concentrations of ethanol solution is shown in Figure 6.

TWA exposure values were estimated based on the measured and modeled concentrations in each exposure scenario (Table 1). *C*_111_ represents the point where the product was used, and *C*_222_ represents the farthest compartment, which is expected to have the lowest concentration among them. The discrepancy among the compartments at the short-term exposure (0.1 h) existed up to 40-fold in scenario 1, and to approximately 10-fold in scenario 2. The deviation of TWA among the compartments decreased after 0.3 h in both scenarios. Additionally, the well-mixed modeling underestimated the TWA by a factor of 1.7 compared to *C*_111_ in the eight-compartment measurement.

## 4. Discussion

Spatial uncertainty was evaluated by simulating the concentration of the eight compartments. Overall, the CV of both exposure scenarios in this study was less than 0.5 within 20 min, while higher variations appeared at the short-term exposure. The initial exposure time during which the spatial uncertainty is significant may differ depending on the conditions of the room. In this model, *λ_ex_* is an indicator that represents the mixing of air in the chamber by convection and dispersion. As shown in Figure 3, increasing the *λ_ex_/**λ* value, for example with a thermal gradient using an air conditioner or heater to increase the convectional flow, can shorten the time required for almost complete mixing. Additionally, human occupants introduce a thermal gradient that contributes to a higher spatial uncertainty [27,28].

The most plausible factor contributing to the spatial uncertainty is the generation of aerosols from the spray. Deposition of heavier aerosols on the floor occurs easily, and a phase transition into the gaseous phase is necessary for suspended VOC aerosols. A dust cleaner produces less aerosols (Appendix A) and thus its use is appropriate for discovering the effect of phase transition on the uncertainty. As shown in Figure 3b, the spatial uncertainty of exposure scenario 2 using a propellant spray decreases due to convection, dispersion, and air mixing through the air injected into the chamber. However, in exposure scenario 1 using a trigger-type spray, the phase transition from liquid aerosols to vapor occurs after the application of the product, and eventually appears to have greater spatial uncertainty during the initial stage (<0.10 h) compared to exposure scenario 2. Based on the results of this study, VOCs must undergo a phase transition to a gaseous state when applied as liquid aerosols. Several studies using engineered nanoparticle (ENP) sprays (e.g., silver nanoparticles) observed that the spray nozzle types and size distribution of aerosols are important for estimating the inhalation exposure [29,30,31,32,33], although VOCs are expected to vaporize faster and achieve even distribution more easily than ENPs in spray products. The results of this study suggest that the estimation of inhalation exposure to VOCs should consider the spatial uncertainty, particularly regarding short-term exposure.

The composition of an active chemical in the spray content was also expected to contribute to the spatial uncertainty. Most VOCs in consumer products are mixed with other components such as solvents and surfactants. The fate of VOCs could be affected by the properties of the mixture. For instance, the vapor pressure of ethanol differs with its mole fraction in a water–ethanol mixture [34]. A larger portion of ethanol in the solution is expected to contribute to a more even distribution due to the higher vapor pressure. However, the uncertainty did not significantly vary among the three different ethanol solutions used in this study. The vapor pressure of ethanol in each solution is not markedly different. The volume fraction of ethanol in water is not linearly proportional to the vapor pressure of ethanol, as the mixture demonstrates a non-ideal behavior. The vapor pressure of ethanol from 20 to 80% solution ranges from 5.7 to 7.6 kPa at 25 °C [34]. Thus, the increase in ethanol composition showed less correlation with the spatial uncertainty.

Other factors such as the structure of a room may affect the spatial uncertainty. Studies using CFD techniques have estimated that articles in the room are likely to affect the spatial distribution. For instance, furniture and walls act as obstacles for the well-mixing condition [27,35]. They affect the homogeneity by hampering the convection along with absorbing the chemical from the indoor air. The ventilation system of the chamber also affects the well-mixing condition. Differences in the opening locations for the inflow and outflow may affect the ventilation efficiency of the chamber [35,36].

Future study is suggested to conduct with other exposure scenarios. As this study conducted two exposure scenarios of spray products in which active substances are instantaneously mixed after application, the exposure scenarios of the products emitting VOCs constantly to the air can be conducted. For example, a naphthalene deodorant ball emits constantly and may be distributed unevenly due to the ventilation flow or the location of the product [37]. The evaporated substance may reach the steady state in a room after long-term use of the product. Partitioning properties are also important in the indoor distribution of substances with high octanol–air partition coefficient. [38]

The results of this study show that spatial uncertainty during short-term exposure is shown to be important when comparing the TWA among compartments. The TWA exposure concentration in the breathing zone of the spray user could deviate from those in other test chamber compartments. The short-term deviation of TWA in this study may provide scientific grounds to investigators monitoring indoor chemicals when uneven distribution of chemicals was observed. The type of spray also affected the short-term spatial uncertainty, as the different size distributions of generated aerosols depend on the spray type. For considering the spatial uncertainty, application of an additional safety factor for liquid aerosol sprays used for a short time could be included in the exposure assessment of these products.

## 5. Conclusions

In this study, the spatial uncertainty of exposure to VOCs during the use of spray products was simulated in a room divided by eight compartments. Measurement and mathematical exposure modeling both resulted in large spatial uncertainty in the short-term exposure. Exposure scenarios using different type of spray showed their contribution to the spatial uncertainty. Thus, this study suggests that the spatial uncertainty due to spray products requires consideration for a short-term exposure assessment. Because only two exposure scenarios were evaluated in this study, future research using more possible scenarios is needed for the implementation of the additional assessment factor with more specific and practical considerations.

## Figures and Tables

**Figure 1 ijerph-18-05334-f001:**
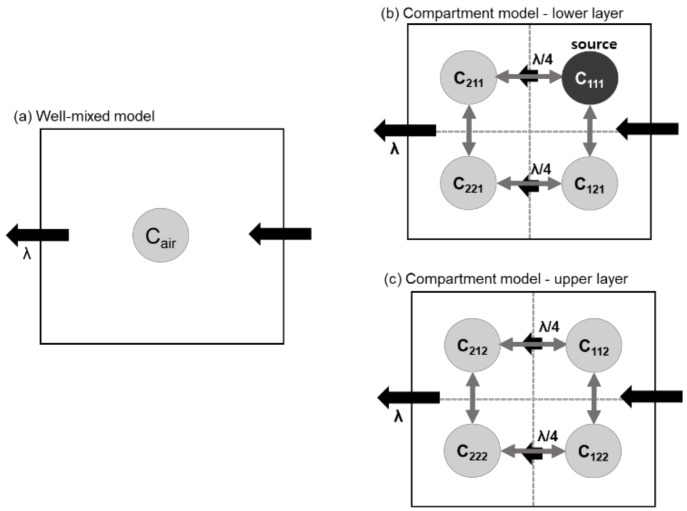
The schematic design of (**a**) model 1 which assumed a well-mixed condition and (**b**,**c**) model 2 which divided the space into eight compartments for the validation of uncertainty. *C_air_* and C_n_ are atmospheric concentrations in a room in model 1 and in each compartment (*n*) in model 2, respectively. λ indicates the total air exchange rate in the room (λ). The exchange rate (*λ_ex_*) between the vertical compartments was not depicted in (**b**,**c**) but was considered in equation 2.

**Figure 2 ijerph-18-05334-f002:**
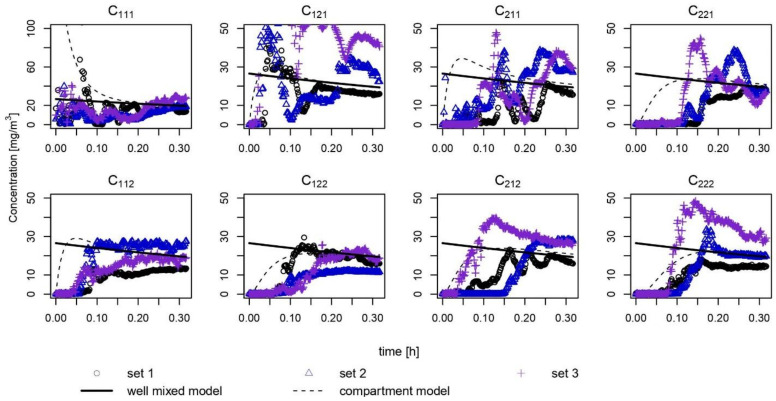
Simulation of a trigger spray containing 50% ethanol solution in the room chamber. The concentration of each compartment was measured in real-time. The scan speed of the instrument was 100 ms (millisecond).

**Figure 3 ijerph-18-05334-f003:**
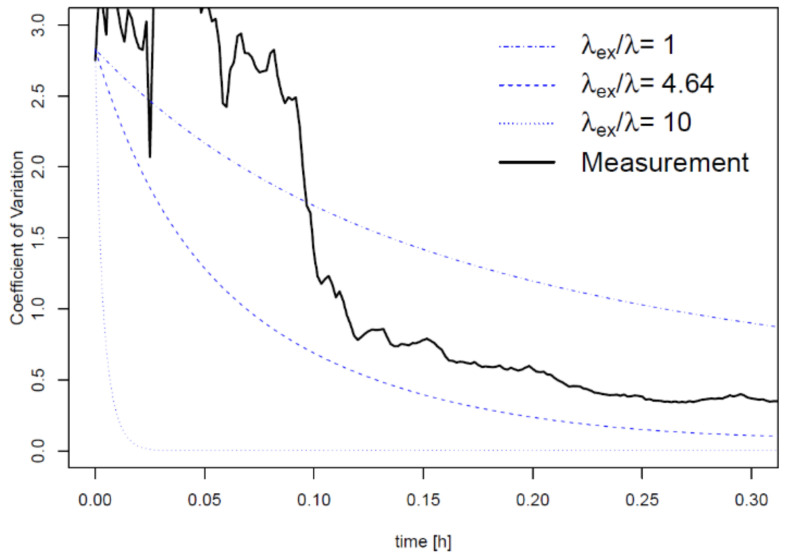
Coefficient of variation (CV) by time using the eight-compartment measurement data in a room-sized chamber (black line) and R simulation (blue lines). λ is the total air flow rate (h^−1^) and *λ_ex_* is the air exchange rate between the adjacent compartments (h^−1^). The CV line at *λ_ex_*/λ is 4.64 is when *λ_ex_* was fitted with the triplicate measurement data.

**Figure 4 ijerph-18-05334-f004:**
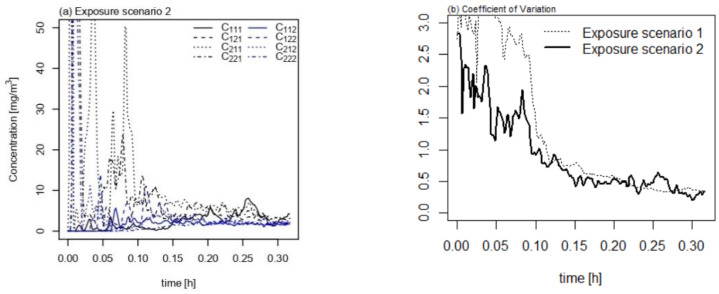
Measurement of (**a**) simulated exposure scenario 2, a propellant spray containing *n*-butane, and (**b**) its comparison to exposure scenario 1, a trigger spray containing ethanol.

**Figure 5 ijerph-18-05334-f005:**
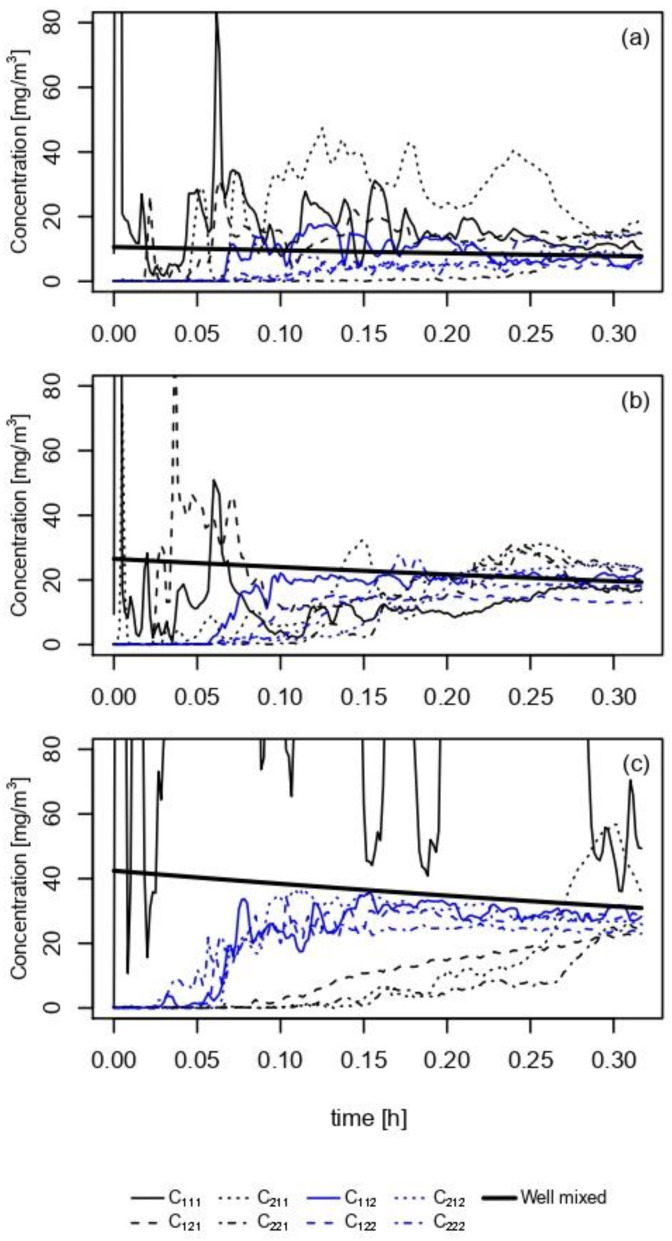
Measurement of exposure scenario 1, a trigger spray containing ethanol mixture at (**a**) 20%, (**b**) 50%, and (**c**) 80% of the volume ratio.

**Figure 6 ijerph-18-05334-f006:**
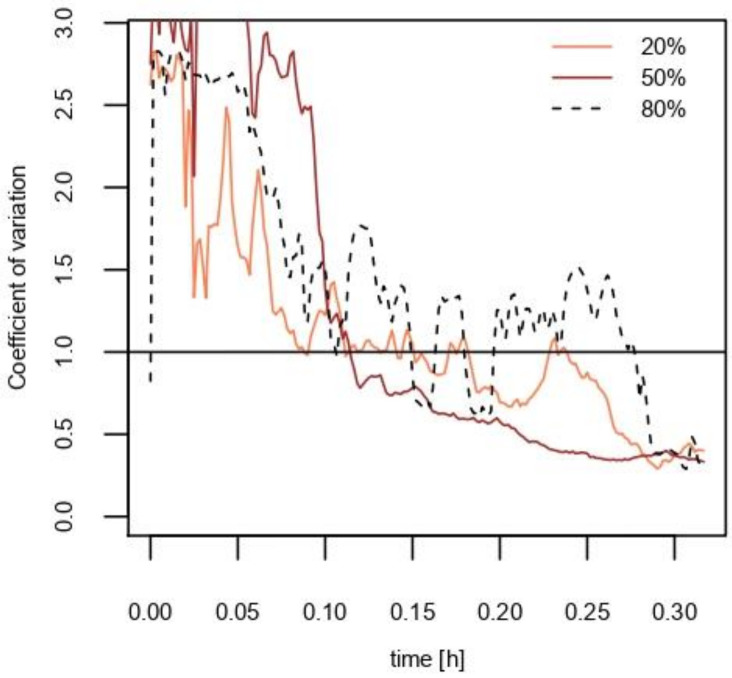
Measurement of exposure scenario 1, a trigger spray containing ethanol mixtures’ coefficient of variation (CV) trends with time.

**Table 1 ijerph-18-05334-t001:** Time-weighted average (TWA) exposure values from the two models using two exposure scenarios with an ethanol trigger (scenario 1) and *n*-butane propellant spray (scenario 2).

Spray Type	Chemical	Exposure Duration (h)	TWA Exposure (mg m^−3^)			
			Real-Time Measurement in Compartments			Well-Mixed Modeling
			*C* _111_	*C* _112_	*C* _222_	
Trigger	Ethanol	0.1	40.1	4.9	1.7	25.2
		0.3	22.3	13.4	16.1	22.9
Propellant	*n*-butane	0.1	0.91	1.0	10.3	-
		0.3	2.3	1.7	4.5	-

## Data Availability

The data presented in this study are available on request.

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
