# Peer review of "Spatial Uncertainty in Modeling Inhalation Exposure to Volatile Organic Compounds in Response to the Application of Consumer Spray Products"

_ijerph, 2021, doi:10.3390/ijerph18105334_

Round 1

Reviewer 1 Report

In the manuscript ID: ijerph-1207552, entitled “Spatial uncertainty in modeling inhalation exposure to volatile organic compounds in response to the application of consumer spray products” the authors evaluate the spatial uncertainty of exposure to VOCs during the use of spray products. The manuscript is well written and of interest but, in my opinion, some parts of the text should be integrated.

Specific comments are reported below:

ABSTRACT

- Line 20: I suggest giving more details about the instrumentation used.

INTRODUCTION

- Line 34: I would add a few more references for the effects of VOCs on human health.

- Line 41: I suggest adding a quote.

- Line 43: I suggest citing some recent studies that have used mathematical models in this field.

- Line 59: Can the authors add something about user variability as well?

MATERIALS AND METHODS

- Line 89: Please correct the punctuation (. (Figure S1a)).

- Line 92: Anything more can be said about temperature and relative humidity? Have they been kept constant? What were the values?

- In my opinion, it is not clear how the study was conducted: how many replicas were made? what was the number (N) of data? I recommend better detail these aspects.

RESULTS

- Figure 2: In the caption, since it was specified that real-time techniques were used, I recommend specifying the acquisition rate of the instrument.

- Figure 3: Why was the top of the graph cut off? I would like to report the graph in full (as well as for the graphs in figure 4, etc.). If the authors think it is not useful, I still recommend specifying the reason for the choice.

- Figure 4b. I recommend specifying in the caption what the vertical blue dotted line means.

DISCUSSIONS

- If I am not mistaken, figure S2 is never mentioned in the text: I suggest you refer to the graph also in the main text.

- Did the authors find any limitations in this study design? In this case, I suggest specifying it in the discussions. Likewise, I suggest (better) reporting the benefits this study can bring to the scientific community.

Author Response

ABSTRACT

- Line 20: I suggest giving more details about the instrumentation used.

:The instrument for real-time measurement is explained in the abstract by adding “by proton transfer reaction mass spectrometry” (Line 21).

INTRODUCTION

- Line 34: I would add a few more references for the effects of VOCs on human health.

: Couple of references from toxicology and epidemiology field are added to support the effects of VOCs on human. (Line 37, 41) Potential neurotoxic effects were implied by rat experiment. Also, an epidemiological study in Argentina supports the claim that VOCs may cause asthma.

- Line 41: I suggest adding a quote.

: “requested by manufacturers and importers” is substituted by “several national regulatory authorities” to clarify information from the newly referred citation. (Line 44)

- Line 43: I suggest citing some recent studies that have used mathematical models in this field.

: Recent studies (published in 2017 and 2020) using mathematical models were cited. (Line 48)

- Line 59: Can the authors add something about user variability as well?

: Reference 9 (Milner et al., 2001) explained that the occupant behavior can be an important information for model exposure, so “occupant behavior” is added after the other demanding inputs. (Line 63)

MATERIALS AND METHODS

- Line 89: Please correct the punctuation (. (Figure S1a)).

: The punctuation was corrected. ((Figure S1a). )

- Line 92: Anything more can be said about temperature and relative humidity? Have they been kept constant? What were the values?

: The range of the relative humidity and temperature of the room chamber during the observations is added (Line 96). It was observed that they had been kept within those ranges.

- In my opinion, it is not clear how the study was conducted: how many replicas were made? what was the number (N) of data? I recommend better detail these aspects.

: The number of observations for each exposure scenario is added. (Line 172). The additional sentence is as follows: “Eight compartments of the room chamber were monitored triplicate for the exposure scenario 1 and once for the exposure scenario 2”.

RESULTS

- Figure 2: In the caption, since it was specified that real-time techniques were used, I recommend specifying the acquisition rate of the instrument.

: The acquisition rate of the instrument is added in the Figure 2 caption. The additional sentence is as follows: The scan speed of the instrument was 100 ms (millisecond).

- Figure 3: Why was the top of the graph cut off? I would like to report the graph in full (as well as for the graphs in figure 4, etc.). If the authors think it is not useful, I still recommend specifying the reason for the choice.

: The previous plot with its smaller y-axis was not meaningful, so it is now substituted with a plot with the same y-axis scale as in Figure 4.

- Figure 4b. I recommend specifying in the caption what the vertical blue dotted line means.

: The dotted line was intended to show the time point (= 0.1 h) when CV values of each exposure scenario seems to be similar but not based on any statistical process. The vertical blue dotted line in Figure 4 is eliminated because the plot still explains well about what was intended.

DISCUSSIONS

- If I am not mistaken, figure S2 is never mentioned in the text: I suggest you refer to the graph also in the main text.

: The figures are now referred in line 116 with a specific and related explanation.

- Did the authors find any limitations in this study design? In this case, I suggest specifying it in the discussions. Likewise, I suggest (better) reporting the benefits this study can bring to the scientific community.

The limitation of this study is added as the suggestion for the future study. (from Line 306) The benefits of this study are summarized in the last paragraph of the discussion part. (from Line 317)

Reviewer 2 Report

I have added comments on the articles.

It will be worth mentioning the study design in the abstract session. In the method a full description of the monitoring instrument is required. The location of the samplers in respect to the source of release and compartment under consideration. The author should define how the near and far field was chosen (was this based on the distance from the source? If so, please indicate the distance between the two fields). Was there background measurements taken before the actual monitoring? The description of the exposure scenario in the method section needs to be improved. For example, the early stage of exposure must be time based.

In the results section, there is an indication that aerosols were not detected in the case of a propellant is less convincing. Hence, I asked if the background measurements were taken. Moreover, information on the number of measurements and the duration of the exposure scenarios needs to move to the methodology part.

Author Response

It will be worth mentioning the study design in the abstract session. In the method a full description of the monitoring instrument is required.

: The main points of the study design in this study were 1) to conduct compartment modeling and 2) to compare simulation results in room-sized chamber and from solving mass balance equations. Though those are included in the abstract section, the name of the real-time analyzer is added to clarify the measuring method. Its brief description of mechanism is added in the method section as well. (Line 157).

The location of the samplers in respect to the source of release and compartment under consideration. The author should define how the near and far field was chosen (was this based on the distance from the source? If so, please indicate the distance between the two fields).

: The description regarding the compartments in the room-sized chamber is added in Line 169-172. The compartments were divided to provide equal volume and one of them was chosen for the source compartment. The distance between the nearest and farthest two locations is added in the main text (1.2 m and 1.95 m, respectively).

Was there background measurements taken before the actual monitoring? The description of the exposure scenario in the method section needs to be improved. For example, the early stage of exposure must be time based.

: The specific explanation of background concentration measurement is added in the method section. (Line 176 and 186) The background measurements were all taken before each observation for at least 5 min, or even more if the ventilation of the room chamber after the previous observation was not enough to reach the baseline concentration. The background concentration was subtracted from the measured concentration during monitoring to eliminate the effect from the deviation of each background concentration. The “early stage of exposure” was all substituted to “short-term exposure” which has been widely used.

In the results section, there is an indication that aerosols were not detected in the case of a propellant is less convincing. Hence, I asked if the background measurements were taken.

: The explanation of aerosol size measurement is added in the method section. (Line 116) The size distribution of background (before the application) and the product (after the application) were measured using an optical particle sizer (OPS) and the results are described in Figures S2 and S3.

Moreover, information on the number of measurements and the duration of the exposure scenarios needs to move to the methodology part.

: The number of measurements is added in Line 172. The duration of the exposure scenarios was already in the method section, but a more detailed sentence is added to illustrate the duration of background measurement. (Line 176).

-Line 19: What is the exposure space?

: For better understanding, it is changed into “a room for simulation” (Line 19)

-Line 20: What about ethanol trigger and propellant. Please correct.

: To clarify the sentence, “consumer spray products” which used to be before “an ethanol trigger and a propellant) was deleted. (Line 20)

-Line 20: What was used for real-time measurements?

: The instrument for real-time measurement is explained in the abstract by adding “by proton transfer reaction mass spectrometry (PTR-MS)” (Line 21).

- Line 23: Better be indicated on a time basis. Early stage of exposure is not very clear.

: “Early stage of exposure” in this paper is all converted into “short-term exposure” which is more widely accepted. (Line 24, 27, 201, 222, 250, 261, 285, 314, 317, 327, and 330)

-Line 102: Provide more details regarding the description of the real-time monitor used in the study.

: As the description and conditions of the PTR-qMS was explained in the later method section 2.5, the sentence informing them in the later section is added. (Line 101) Also, a brief description of how the instrument works is added. (Line 157)

- Line 109: How long was the monitoring carried out for?

: In the section 2.5, the time of the monitoring process was explained that it was conducted for 20 min after the application of the product. Thanks to your comment about the background measurement, the time for the background measurement before the application is added in Line 176. As the background measurement was conducted for at least 5 min, the whole monitoring process for each observation took at least 25 min.

Round 2

Reviewer 2 Report

I am happy with the corrections.